# Design of Hydrogel Microneedle Arrays for Physiology Monitoring of Farm Animals

**DOI:** 10.3390/mi16091015

**Published:** 2025-08-31

**Authors:** Laurabelle Gautier, Sandra Wiart-Letort, Alexandra Massé, Caroline Xavier, Lorraine Novais-Gameiro, Antoine Hoang, Marie Escudé, Ilaria Sorrentino, Muriel Bonnet, Florence Gondret, Claire Verplanck, Isabelle Texier

**Affiliations:** 1CEA, LETI-DTBS, Université Grenoble Alpes, F-38054 Grenoble, France; laurabelle.g@gmail.com (L.G.); antoine.hoang@cea.fr (A.H.); marie.escude@cea.fr (M.E.); ilaria.sorrentino@cea.fr (I.S.); claire.verplanck@cea.fr (C.V.); 2PEGASE, INRAE, Institut Agro, F-35590 Saint-Gilles, France; sandra.wiart-letort@inrae.fr (S.W.-L.); c.xavier@groupe-esa.com (C.X.); florence.gondret@inrae.fr (F.G.); 3INRAE, Université Clermont Auvergne, Vetagro Sup, UMRH, F-63122 Saint-Genès-Champanelle, France; alexandra.masse@inrae.fr (A.M.); muriel.bonnet@inrae.fr (M.B.); 4Plateforme CICS (Centre Imagerie Cellulaire Santé), Université Clermont Auvergne, F-63000 Clermont-Ferrand, France; lorraine.novais-gameiro@uca.fr

**Keywords:** microneedles, animal health and welfare, hydrogel, interstitial fluid analysis, livestock, skin perforation

## Abstract

For monitoring animal adaptation when facing environmental challenges, and more specifically when addressing the impacts of global warming—particularly responses to heat stress and short-term fluctuations in osmotic regulations in the different organs influencing animal physiology—there is an increasing demand for digital tools to understand and monitor a range of biomarkers. Microneedle arrays (MNAs) have recently emerged as promising devices minimally invasively penetrating human skin to access dermal interstitial fluid (ISF) to monitor deviations in physiology and consequences on health. The ISF is a blood filtrate where the concentrations of ions, low molecular weight metabolites (<70 kDa), hormones, and drugs, often closely correlate with those in blood. However, anatomical skin differences between human and farm animals, especially large animals, as well as divergent tolerances of such devices among species with behavior specificities, motivate new MNA designs. We addressed technological challenges to design higher microneedles for farm animal (pigs and cattle) measurements. We designed microneedle arrays composed of 37 microneedles, each 2.8 mm in height, using dextran-methacrylate, a photo-crosslinked biocompatible biopolymer-based hydrogel. The arrays were characterized geometrically and mechanically. Their abilities to perforate pig and cow skin were demonstrated through histological analysis. The MNAs successfully absorbed approximately 10 µL of fluid within 3 h of application.

## 1. Introduction

Improving animal adaptation to environmental challenges in order to ensure synergy between production, health, and welfare is widely recognized as a critical component for the sustainability of livestock production systems. It is also essential for addressing ethical and societal demands aimed at reducing antibiotic use and strengthening consumer confidence. Future environmental conditions are expected to become increasingly variable due to the impacts of global warming, amplifying the challenges animals are facing—such as variability in feed nutritional value, water scarcity, heat stress, and increased pathogen pressure. This underscores the urgent need for robust tools to accurately assess animal adaptation, guaranteeing health and welfare and inform management decisions that facilitate the adoption of novel practices [1].

Timely, non-invasive, precise, and continuous monitoring of physiological traits is therefore an emerging field in farm animal research to better understand the biological pathways underlying adaptation and to effectively manage animal performance, health, and welfare [2]. Consequently, there is a pressing need for minimally invasive, real-time monitoring of key biomarkers in farm animals. These biomarkers may be proxies of other physiological variables or indicate stress, dehydration, or immune status, and serve as control metrics—such as the quantification of antibiotics or hormones [3,4,5,6].

In human medicine, biomarker monitoring via sampling and analysis of interstitial fluid (ISF) has emerged in recent years as a minimally invasive alternative to traditional blood tests [7,8,9,10]. The ISF—a plasma filtrate bathing tissues such as the dermis—contains glucose, lactate, electrolytes, and cytokines, with concentrations that can be correlated with those in blood [7,11,12,13]. Unlike conventional venipuncture, ISF can be accessed using microneedle (MN)-based devices that penetrate the skin with minimal discomfort, enabling continuous, real-time biomarker monitoring over extended periods (hours to days) [7,8,9,10,14].

Microneedle (MN)-based devices, comprising microscopic structures designed to locally breach the outer layers of the skin and access the ISF-rich dermal compartment, have predominantly been developed for human applications, with preclinical validation often conducted in rodent and porcine models. More recently, veterinary applications of MN technologies have emerged, especially for transdermal delivery of active compounds [15,16] and vaccines [17,18] across diverse species (dog, pig, guinea pig). Nevertheless, to the extent of our knowledge, only few studies have explored the use of an MN-based device for ISF sampling and analysis in animals [19,20]. Taylor et al. used commercial stainless-steel hollow MNs assembled in an array through a 3D printed piece to puncture rat skin and uptake ISF [20]. Steinbach et al. reported 1.4 mm high stainless-steel solid microneedles coated by calcium-crosslinked alginate hydrogel for ISF sampling of BCG vaccinated cows to assess the animal response to tuberculin [19].

A few fully integrated MN-based devices have been developed for human measurement of key physiological parameters—such as ions and selected metabolites (e.g., glucose)—using optical or electrochemical sensing [21,22,23]. However, the detection of multiple metabolites, of metabolites requiring more complex analyses (e.g., ELISA), or those present at low concentrations in ISF, demands an alternative approach. In such cases, microneedle arrays (MNAs) can serve primarily as minimally invasive, user-friendly, and painless tools for ISF extraction, which are subsequently subjected to conventional laboratory assays, including liquid chromatography techniques (HPLC, UPLC) or mass spectrometry [8,9,10,24,25].

Microneedles arrays (MNAs) composed of crosslinked, water-insoluble hydrogels, which initially exhibit high mechanical strength to facilitate skin penetration and rapidly achieve significant swelling within minutes, have been reported to harvest substantial volumes of ISF [24,25,26]. In human applications, these systems have been utilized for drug monitoring [24,26] and sampling circulating nucleic acids [27]. However, translating this technology to farm (large) animals (e.g., pigs, cattle) presents additional challenges, primarily due to their mechanically more resistant skin. Increased skin Young’s modulus values (e.g., 2.1 MPa for pig dermis/epidermis [28], 83–170 MPa for films of collagen extracted from cattle’s skin [29]) in comparison to humans (1.6 MPa for dermis/epidermis [28]) complicates MN insertion without compromising device integrity. Furthermore, longer MNs are required to reach the dermis, given the augmented thickness of the epidermis and stratum corneum layers (e.g., ≈55–90 µm for pig [30,31,32], 600–1400 µm for ear cow [33], in comparison to 50–60 µm for humans ([30,31]) and full skin (e.g., 1.3 (ear)–3.6 (back) mm for pig [30,31,32], 1.4–3.2 mm for ear cow [33], 3–8 mm for other cow regions [34,35], in comparison to 1.5 mm for human forearm ([30,31]). Hydrogel-based MNAs are typically produced via molding processes, and the manufacture of devices with taller MNs introduces significant challenges, with risks of mechanical failure during demolding. Moreover, tall MNs are more fragile and could easily break or buckle during animal skin insertion.

This study had two objectives: (1) address the technological challenges involved in fabricating hydrogel-based microneedle (MN) arrays with tall, mechanically robust microneedles using a polymer molding and evaporation process, and (2) establish the proof of concept that these hydrogel-based MN arrays can effectively perforate farm animal skin for the purpose of sampling interstitial fluid (ISF) as a means of monitoring animal physiology. Specifically, we successfully designed and fabricated arrays with MNs measuring 2.8 mm in height, using a crosslinked dextran-methacrylate (Dex-MA) hydrogel. Dex-MA with a 32% degree of methacrylate substitution (DS) was selected for its biocompatibility and its optimized balance between mechanical strength in the dry state (Young’s modulus of 2.5–3 MPa) and swelling capacity (50% swelling ratio at equilibrium when fully immerged in phosphate saline buffer), as demonstrated in our previous study [36]. Dex-MA-based MN arrays were successfully demonstrated to extract fluid from an in vitro hydrogel model mimicking the skin, and to perforate porcine and cow skin samples.

## 2. Materials and Methods

### 2.1. Dextran-Methacrylate (Dex-MA) Synthesis

Dextran-methacrylate with a methacrylate degree of substitution of 32% was synthetized according to the previously described protocol [36]. Briefly, dextran T70 (5 g, Mw = 70,000 g/mol, Pharmacosmos (Holbæk, Denmark)) was dissolved in 100 mL of distilled water in a beaker. After complete dissolution, 3.66 mL of methacrylic anhydride (Sigma-Aldrich, Saint-Quentin Fallavier, France) was added dropwise into the polymer solution. The solution was stirred at room temperature (RT) for 1 h. The pH was adjusted using NaOH (3 mol/L) to keep the pH at around 9–11 during the reaction. The obtained Dex-MA solution was dialyzed against distilled water for 1 week (12–14 kDa molecular weight cut-off membrane, Carl Roth, Lauterbourg, France), and lyophilized. The white solid was stored at −20 °C prior to ^1^H nuclear magnetic resonance (NMR) spectroscopy characterization and use. The ^1^H NMR spectrum in D_2_O at 300 MHz, δ (ppm) was as follows: 2.1 (s, 3H, CH_3_-methacrylate), 3.5–4.2 (m, 6H, H-2, H-3, H-4, H-5, H-6, H-6′), 5.1–5.4 (m, 1H, H-1), 5.9 (m, 1H, H-vinyl), 6.5 (m, 1H, H-vinyl).

### 2.2. Microneedle Array Fabrication

An aluminum master mold featuring the targeted geometry (Figure 1)—an 18 mm diameter microneedle array (MNA) comprising 37 pencil-shaped MNs with a base diameter of 500 µM, each 2800 µm in length, and spaced 1400 µm from tip to tip—was fabricated via precision micromachining. Subsequently, a polydimethylsiloxane (PDMS) mold was produced by casting a degassed PDMS precursor solution with a 10:1 mass ratio of base to crosslinker (Sylgard 184 PDMS kit, Sigma-Aldrich, Saint-Quentin Fallavier, France). The cast was cured at 80 °C for 3 h, followed by manual demolding from the aluminum master.

The polymer solution was prepared in a round-bottom flask covered with aluminum foil, consisting of Milli-Q water containing 20% (*w*/*v*) Dex-MA (DS = 32%), 1% (*w*/*v*) lithium phenyl-2,4,6-trimethylbenzoylphosphinate (LAP, Sigma-Aldrich, Saint-Quentin Fallavier, France) as a photo-initiator, and 5% (*w*/*v*) sorbitol (Sigma-Aldrich, Saint-Quentin Fallavier, France) as a humectant. The solution also optionally contained trace amounts of dye (Neutral Red, Brillant Blue) to enhance visualization of the MNs, particularly during phantom and animal skin perforation experiments. The solution was stirred under vacuum for at least 2 h to ensure homogeneity and degassing. It was then cast into the PDMS mold (associated with the demolding piece), which was mounted on a steel support connected to a vacuum pump. A reduced pressure of 50 mbar was applied for 30 min to facilitate complete filling of the MN tips.

The polymer solution was dried within the mold at 35 °C for 48 h. Prior to demolding the MNAs, the Dex-MA hydrogel was crosslinked via photopolymerization by first irradiating the baseplate of the array with 405 nm light (75 mW/cm^2^) for 5 min. Following demolding, the MN side of the patch was subjected to a second 5 min irradiation under identical conditions to ensure complete crosslinking.

Digital microscopy images of the aluminum master and MNAs were acquired with a Keyence VHX-7000 digital optical microscope (Keyence, Osaka, Japan) in reflectance mode with ring illumination.

### 2.3. Mechanical Characterization of the Microneedle Arrays

Mechanical testing of the MNAs was performed with a TA.XT Plus texturometer (StableMicrosystems, Surrey, GU7 1YL, UK). The MNAs were affixed to a circular probe of 18 mm diameter (matching the MNA size) with the MNs oriented downward. Their mechanical resistance was characterized by uniaxial compression tests against a stainless-steel plate with a maximum compressive load of 49 N (maximum capacity of the device equipped with 5 kg force cell) at a speed of 0.1 mm/s. Following loading, the instrument executed an unloading phase and then returned to its initial position.

### 2.4. Model Skin Perforation Experiments

SynTissue adult skin (2N) (SynDaver, Tampa, FL, USA) was used as a commercial model of human skin. SynTissue pieces (cut to 3 × 7 cm dimensions) were set on a custom-designed test bench that allowed the tensing of tissue for the perforation assays. Additionally, a rigid backing support was placed beneath the synthetic skin to provide consistent counter-resistance during insertion. An Orbit Inserter (MyLife, Burgdorf, Switzerland) was used as an applicator tool to reproducibly insert the MNAs with a suitable speed and force into the SynTissue model set on the perforation bench. To attach the MNA to the Orbit Inserter, an adaptor piece was designed and manufactured by 3D printing and the MNA was affixed onto it using double-sided tape. After the MNA was applied on the SynTissue, a constant pressure was maintained for 30 s before the applicator was removed, in order to favor reproducible tissue perforation and not only deformation. The MNA, still attached to the adaptor piece, remained in the skin for 2 additional minutes before being removed. The SynTissue and MN devices were then characterized by digital microscopy.

### 2.5. Model Skin Fluid Uptake Experiments

A 20 mm thick gelatin/agar hydrogel, mimicking the dermis, was prepared by mixing 70 g of gelatin (type A, gel strength 300, Sigma-Aldrich), 2.9 g of agar (Sigma-Aldrich), and 10 mg of neutral red dye (to ease visualization) in 290 mL of PBS (phosphate buffer saline, 10 mM phosphate, 137 mM NaCl, 2.7 mM KCl, pH 7.4). The mix was heated in a microwave for 2.5 min at 350 W, energetically stirred with a spatula, and rapidly poured into a 13.6 cm diameter Petri dish. The hydrogel was covered with an aluminum foil to provide a non-humid surface to the skin phantom. The dry MNAs were weighted, then applied onto this skin model using the Orbit Inserter tool (MyLife, Switzerland), and maintained into the phantom for 15 min, 3 h, or 24 h before careful removal. After MN device retrieval, MNAs were weighted and observed with a Keyence VHX-7000 digital optical microscope; the gelatin/agar hydrogel was also observed by digital optical microscopy.

### 2.6. Ex Vivo Animal Skin Perforation Experiments

For ex vivo pig experiments, a conventional growing pig (crossbred Pietrain × (Large White × Landrace) weighing about 50 kg (about 80 days of age)), humanely killed after electronarcosis at the experimental slaughterhouse of the UE3P unit (Pig Physiology and Phenotyping Experimental Facility, INRAE, 35 590 St-Gilles, France, http://doi.org/10.15454/1.5573932732039927E12, accessed on 1 July 2025), was used. For cow skin experiments, a Limousine cow (weighing 775 kg, approx. 11 years old) was killed at the experimental slaughterhouse of Herbipôle Research Unit (INRAE, 63 122 Theix, France, http://doi.org/10.15454/1.5572318050509348E12, accessed on 1 July 2025), and desired skin samples were uptaken. The perforation experiments were conducted immediately after the death of the animals, to avoid modifications of skin texture.

The MNAs were applied onto various locations of the animal skin using either the Orbit Inserter applicator or manual thumb pressure. A consistent pressure was maintained for 30 s during application. Following insertion, the MNAs remained in place onto the skin for approximately 15–20 min for pig experiments, 1 min for cow experiments, before removal. After application, the MNAs were carefully retrieved and examined using a Keyence VHX-7000 digital optical microscope.

The margins of the tissue to be collected were marked around the MNA application area. A scalpel was then used to carefully excise the surrounding tissue and isolate the sample. The tissue samples were bisected through the center of the MNA application site to allow proper placement into histology cassettes. Immediately after sectioning, the samples were immersed in 4% paraformaldehyde (Thermo Fisher Scientific, Cat. N°: J19943.K2, Waltham, MA, USA) at room temperature for 4 h. Following fixation, the samples were stored at 4 °C until transfer to UMR Pegase for pigs; for cattle, samples were stored at UMR Herbivores and then the Centre Imagerie Cellulaire Santé. Paraffin embedding, microtome sectioning, hematoxylin-eosin (H&E) staining, slide mounting, and microscope imaging were carried out to perform tissue perforation measurements.

## 3. Results

### 3.1. Microneedle Arrays (MNA) Design and Fabrication

Dextran-methacrylate (Dex-MA)-based MNAs were fabricated according to a molding-evaporation process according to the classical literature procedures [36], as described in Figure 2a. An aluminum master comprising 37 pencil-shaped microneedles with a center-to-center spacing (pitch) of 1.4 mm, a height of 2.8 mm, and a base diameter of 500 µm was machined and subsequently used to fabricate polydimethylsiloxane (PDMS) negative molds.

The Dex-MA solution was then poured into the molds that were filled until the MN tips with the aid of a vacuum. The polymer solution was thermally dried, and UV-crosslinking of methacrylate groups was performed by irradiating the top of the device. After demolding, UV-crosslinking was also performed on the MN tip side. However, two challenges were encountered in the fabrication of these high MN devices (2.8 mm height MN) that were not previously observed for shorter ones (<1200 µm) [36].

First, the demolding step could not be performed efficiently with high MN arrays. As displayed in Figure 2b, bending the PDMS mold was shown to be ineffective since it caused damage to both the array baseplate and the MNs because of the high mechanical constraints. Therefore, a poly(methyl methacrylate) (PMMA) demolding piece was designed and fabricated by machining (Figure 2c). It allowed for vertical pulling of the dried MNAs that could be easily demolded without breaking.

Secondly, the MNAs obtained, which were initially based on a polymer solution containing only Dex-MA at 20% *w*/*v* and the LAP photo-initiator at 1% *w*/*v* in distillated water, presented several defects. While the basis of all microneedles could be present on the array after demolding, the majority of the device’s MNs contained internal bubbles that could induce MN fragility and eventually breaking. Moreover, the baseplate of the MNAs was extensively cracked, rendering the devices unusable. Such defects could originate from a poor mold-filling process, insufficient quantity of polymer or excessive solution viscosity, or inappropriate water evaporation during drying. Different optimization assays revealed the issue stemmed from the drying phase. Consequently, drying parameters were modified based on the premise that the previous drying protocol (24 h at 40 °C followed by 24 h at 80 °C) induced excessive drying rates, likely causing rapid water evaporation, material shrinkage, and subsequent cracking and bubble formation. The drying regime was altered to a single-stage drying for 48 h at 35 °C but with only partial improvement in MNA quality, which continued to exhibit significant bubble formation within the microneedles and cracking of the baseplate (Figure 3a). To achieve controlled drying while maintaining reasonable processing times, the incorporation of a humectant additive was therefore proposed. Sorbitol and glycerol, commonly used humectants in pharmaceutical and food formulations, were selected for evaluation based on concentrations reported in the literature [37,38,39]. Formulations containing glycerol did not yield conclusive improvements. In contrast, MNAs prepared with Dex-MA formulations containing 5% *w*/*v* sorbitol exhibited baseplates free of cracks and MNs devoid of bubbles (Figure 3b).

The MNAs obtained with this optimized process were geometrically characterized using digital microscopy and demonstrated reproducible and satisfactory fidelity to the aluminum master mold, particularly with a microneedle height of 2661 ± 15 µm (>95% of targeted value).

### 3.2. MNA Mechanical Characterization

To evaluate the mechanical properties of the Dex-MA MNAs, compression tests were conducted against a stainless-steel plate with a maximum compressive load of 49 N (Figure 4a). The photographs of the devices before and after compression (Figure 4b) showed that all microneedles were present after compression, and structurally unbroken, as also evidenced by the continuous loading and unloading curves recorded by the texturometer that did not show disruption, therefore indicating no MN break (Figure 4c). This result suggested that each microneedle could withstand an applied force of approximately 1.3 N without failure. Only a slight deformation at the MN tips could be observed. Importantly, no MN buckling was evidenced, demonstrating the mechanical resistance of the dried polymer material.

### 3.3. Development of Perforation Protocol on Human Skin Model

A perforation protocol was developed using SynTissue as a commercial human skin phantom and a slightly modified commercial Orbit Inserter applicator (Figure 5a), through the use of a dedicated bench that allowed for the pre-tensing of the synthetic tissue (Figure 5b). The MNAs were inspected before and post-perforation to identify any structural damage incurred during insertion (Figure 5c). The integrity of the Dex-MA MN devices was maintained post-perforation, with all microneedles intact and free from fracture. However, a shear band oriented at approximately 45°—denoted by the white dotted lines in Figure 5c—was occasionally observed in some microneedles following the perforation test, although no material failure occurred along this band. Furthermore, the 37 perforation patterns generated by the MNs into the skin phantom was clearly visible after array insertion (Figure 5d), indicating a 100% perforation rate across all needles. This experiment was replicated multiple times, yielding highly consistent and reproducible outcomes. Collectively, these results confirm the high efficacy of the devices in perforating synthetic human skin phantoms using the developed application protocol.

### 3.4. Fluid Uptake Experiments on Model Skin

SynTissue, a textile impregnated with a water/ethanol preservative solution and featuring a persistently moist surface, does not replicate the water content of biological skin and is therefore unsuitable for fluid uptake studies. To better assess the ISF uptake capacity of the MNAs, we employed a gelatin/agar hydrogel containing approximately 75% water, as a tissue-mimicking model. To provide a dry interface with the MN device, the hydrogel surface was covered with an aluminum foil (Figure 6a). After Dex-MA MNA insertion for 15 min, 3 h, or 24 h, the gelatin/agar hydrogel was examined via digital microscopy, revealing a distinct pattern of 37 MN perforations (Figure 6b). The difference in Dex-MA MNA mass before and after insertion was used to quantify the amount of fluid absorbed during the application period (Table 1). The inclusion of a dye in the hydrogel matrix enabled visual tracking of fluid diffusion into the MN tips, providing qualitative confirmation of uptake (Figure 6c–e; Table 1). We observed a time-dependent increase in both the volume of absorbed fluid and the extent of dye penetration into the MN tips, reaching a maximum uptake of 31.4 mg after 24 h. Notably, the fluid diffused along approximately 70% of the MN length, regardless of insertion duration. It is important to note that after 24 h application, some MNs were broken after removal of the device from the skin phantom.

### 3.5. Porcine Skin Perforation Experiments

Porcine skin perforation experiments were conducted on the euthanized animal. Dex-MA MNAs were applied using the Orbit Inserter at various anatomical sites (Figure 7a): the right side of the neck behind the ear (1), lower (2) and upper (4) regions of the neck, the lateral side of the right forelimb (3), the chest between the forelimbs (5), and the inner surface of the ear (6). No structural damage to the MNAs was observed either during insertion or upon removal of the devices (Figure 7b) after an application time ranging from 10 to 20 min.

The MNAs achieved successful skin perforation in most regions, including the neck, shoulder, forelimb, and chest (Figure 7c), but failed to penetrate the skin of the inner ear (region 6). Histological analysis (Figure 7d,e) revealed penetration depths between approximately 200 and 1000 µm, with the MNs effectively breaching the stratum corneum and reaching the dermal layer.

### 3.6. Cow Skin Perforation Experiments

Ex vivo skin perforation experiments were conducted on freshly excised cow ear and neck tissues obtained immediately post-euthanasia. Complete insertion of the MNAs using the Orbit Inserter (Figure 8a) was not achieved in neither the ear nor the neck tissue. Consequently, successful insertion was accomplished manually via thumb pressure, accompanied by the application of a counterforce (Figure 8b–d). Nevertheless, device adhesion to the ear tissue was suboptimal, whereas the neck region demonstrated qualitatively superior compatibility in terms of both insertion facility and device retention.

Histological analysis of both tissue types confirmed successful MN penetration through the stratum corneum into the dermal layer (Figure 8e–g). Although additional studies are warranted to reinforce these findings, preliminary observations suggest that MN perforation in the neck tissue was generally less deep (276 ± 86 µm) than in the ear (363 ± 227 µm), but more reproducible.

## 4. Discussion

### 4.1. Hydrogel-Based MNA for ISF Uptake

Real-time, minimally invasive monitoring of physiological biomarkers is critical for effective assessment and management of animal performance, health, and welfare, and more importantly, using wearable devices allowing continuous measurements without restraining the animals. This necessitates the development of advanced analytical tools. In human medicine, microneedle arrays have been recognized for their ability to enable painless extraction of dermal interstitial fluid (ISF), a biofluid rich in clinically relevant biomarkers [8,10,11,40]. For applications involving global biomarker profiling or discovery of multi-analyte signatures, ISF sampling followed by advanced analytical techniques—particularly omics-based approaches—represents a promising strategy.

Chemically crosslinked hydrogel-based MNAs are particularly well-suited for this purpose. These hydrogels form three-dimensional polymer networks capable of absorbing large volumes of aqueous fluid without dissolving, thereby maintaining their structural integrity. When dried and molded into MN structures, they possess sufficient mechanical strength to perforate the skin and rapidly absorb ISF upon contact with the dermal compartment [24,25,26].

In this study, we selected photo-crosslinked dextran-methacrylate (Dex-MA) as the material of choice for MNA fabrication. In previous work, we systematically evaluated the mechanical and swelling properties of Dex-MA synthesized from dextran with varying molecular weights and degrees of methacrylate substitution (DS) [36]. We identified Dex-MA derived from dextran of 70,000 g/mol molecular weight and a DS of 32% as an optimal formulation for a MNA dedicated to ISF uptake [41]. This formulation demonstrated a favorable balance between swelling capacity (approximately 50% swelling ratio at equilibrium when fully immersed in phosphate buffer) and mechanical toughness in the dry state, with a Young’s modulus of 2.5–3 MPa [36]. This modulus exceeds that of porcine skin (approximately 2.1 MPa for dermis/epidermis [28]), suggesting sufficient rigidity for tissue penetration. In the present study, dried photo-crosslinked Dex-MA material, when shaped into a microneedle array, withstood compressive forces up to 50 N against a steel plate without exhibiting needle fracture or buckling. The previous literature reports insertion forces ranging from a few Newtons up to 30 N in human or porcine skin [42,43,44], once again suggesting sufficient mechanical resistance to perforate porcine skin.

While specific data for cattle skin are lacking, reported Young’s modulus values for films derived from bovine skin collagen range from 83 to 170 MPa [29]; however, these values could be far above those of native tissue properties. It is also important to note that the Young’s modulus represents only one aspect of the skin’s complex mechanical behavior, which is highly viscoelastic and varies depending on anatomical location.

### 4.2. MNA Design

As with mechanical properties, skin thickness varies markedly across body regions. Therefore, the anatomical site of application significantly influences MNA design, particularly with respect to microneedle height. For example, Simon et al. reported bovine ear skin thickness ranging from 1000 to 1400 µm, depending on the specific area (e.g., apex, base, or top of the pinna) [33], while other anatomical regions in cattle exhibit skin thicknesses between 3 and 8 mm [34,35]. In pigs, skin thickness ranges from approximately 1.3 mm in the ear to 3.6 mm on the back [30,31,32].

In our study, we primarily considered anatomical regions that would be compatible for a sensor worn by farm animals moving in their housing environment. This constraint, together with previous skin thickness measurements in cows [33], have been considered for designing the MNs. The ear presents a seemingly attractive site for MNA application in cattle, given its current use for identification tagging. However, its high cartilage content may hinder effective skin perforation and the ISF (hydration) content of this tissue is unknown. Alternatively, the neck may offer a more suitable site due to its relatively homogeneous tissue composition and accessibility; it also allows for secure post-application strapping to stabilize the device. In pigs, auricular sites are not ideal due to social behaviors of the animals when reared in collective pens, particularly ear-biting and fighting, which could compromise device integrity and longevity.

Considering interspecies and inter-individual variations, as well as the variability in skin properties across anatomical sites, we designed MNAs with a microneedle height of 2.8 mm. This dimension was selected based on available histological data for porcine and bovine skin [30,31,32,33], and with the aim of ensuring consistent dermal penetration and reliable ISF access across diverse tissue types and breeds. Notably, we considered that using arrays presenting tall MNs would offer the most practical and reliable option for conducting animal experiments across different anatomical sites, both in pig and cow.

### 4.3. MNA Fabrication

Fabricating MNAs with 2.8 mm height microneedles and a tip-to-tip spacing of 1400 µm poses considerable technical challenges, particularly during the demolding step, which is highly dependent on array geometry. For humans, all studies describe MNs with a maximum of 1.5–2 mm in height to minimize pain [45,46]. For veterinary applications, studies have described arrays with an MN height of maximum 1500 µm, and those for drug or vaccine delivery often concern MNs around 600–650 µm tall [15,16,17,18,19,20]. As the MN height and density increase, perpendicular removal from the mold becomes essential. For arrays with MNs of 800–1400 µm in height, gentle mold flexing typically suffices to release the array intact. However, for taller MNs as developed here, mold bending caused basal plate fragmentation and MN detachment, rendering the devices nonfunctional. To address this, a custom PMMA demolding insert, matching the 18 mm diameter mold, was fabricated and positioned beneath the hydrogel during casting. It facilitated vertical extraction of the patch, preserving structural integrity.

Initial arrays made from Dex-MA/LAP formulation exhibited internal bubbles and severe baseplate cracking, compromising device quality. Attempts to slow drying by adjusting process parameters were ineffective. To address this, a humectant—sorbitol, widely used in pharmaceutical and food formulations [38,39]—was incorporated to retain water and control drying kinetics. Sorbitol significantly reduced cracking and bubble formation, demonstrating its potential to improve mechanical robustness during MNA fabrication.

These protocol enhancements enabled the successful fabrication of MNAs with 2.8 mm-high MNs—the tallest hydrogel-based microneedles reported to date fabricated through a polymer molding/evaporation process, to the best of our knowledge.

### 4.4. MNA Skin Perforation

With the tall MNAs fabricated via a reliable and reproducible process in hand, we next focused on developing a robust and consistent perforation protocol. The insertion force and application speed of an MNA are critical parameters for efficient skin penetration [43]. To deliver a precisely controlled impact, we selected a commercial Orbit Inserter applicator and engineered an adaptor to securely mount the MNAs onto this platform.

The application protocol was initially optimized using a synthetic skin model, SynTissue, which mimics the dermis (1 mm thickness) and hypodermis (3 mm thickness) layers of human skin, and is typically employed for surgical incision and suture training. Additionally, a custom-designed test bench was utilized to pre-tense the tissue, simulating in vivo conditions. Under these conditions, we achieved 100% perforation efficacy with the MNAs, and the MNs withstood the mechanical stresses of insertion, although early signs of material microfractures were observed in some needles.

Subsequently, ex vivo perforation studies were performed on freshly excised porcine and bovine tissues obtained immediately post-euthanasia. Multiple anatomical sites (ear, neck, shoulder, forelimb, and chest) were assessed for porcine skin perforation. Successful perforation was achieved at all sites except the ear, which we attribute to the underlying cartilage beneath the thin skin layer, complicating patch adhesion. Furthermore, the natural social behaviors of pigs render ear application challenging in practical settings. For bovine tissue, ear and neck sites were tested; however, the Orbit Inserter was ineffective in delivering the MNAs into these tissues. Moreover, partial or complete device breakage could occur during application. Consequently, manual application by thumb pressure with a counterforce applied to the opposite side of the tissue was employed, providing a less reproducible but more adaptable method to ensure perforation. Although the ear was initially considered an optimal site and histology confirmed successful perforation, the neck emerged as a more practical and comfortable location, consistent with observations in porcine skin. It is important to note that Steinbach et al. also used the side of the cow neck to apply their MN-based ISF sampling device to assess the animal response to tuberculin [19].

It is established that MNs typically penetrate approximately 60% of their length into human skin during insertion [47]. In gelatin/agar phantoms, we observed MN penetration depths of approximately 60–75% of their length (1600–1900 µm). However, histological analysis revealed substantially lower perforation depths in porcine skin (200–1000 µm) and bovine skin (139–645 µm). This discrepancy may reflect a reduced penetration efficiency due to intrinsic differences in animal skin properties relative to human, as well as partial tissue collapse during histological processing, complicating accurate quantification. Additionally, prior studies have demonstrated that rigor mortis rapidly develops post-euthanasia, leading to increased porcine skin stiffness that diminishes the MNA penetration depth compared to living tissue [48]. Therefore, the perforation efficiencies measured ex vivo likely underestimate those achievable in vivo.

Nevertheless, these preliminary results demonstrate the capability of Dex-MA-based MNAs to perforate both porcine and bovine skin, validating their potential for further in vivo investigation. Future studies exploring various design parameters (e.g., microneedle shape, length, and density) could also enable greater or more consistent perforation depths.

### 4.5. MNA Fluid Uptake

This study aimed to demonstrate the design feasibility of a Dex-MA hydrogel-based MN array and to select the best anatomical location to achieve skin perforation in large farm animals. Following the rules of 3Rs, the skin samples were obtained post mortem from healthy euthanized animals. Fluid circulation ceases following the loss of cardiac activity. Therefore, we did not intend to assess and analyze fluid absorption in these euthanized animals. Rather, we conducted swelling studies under well-controlled conditions. Thus, a synthetic gelatin/agar phantom was used, which possesses approximately 80% water content—comparable to the 70–75% water content reported for porcine skin [49]. In this model, fluid uptake by the hydrogel MNs was evident, with absorption reaching 10.7 µL after 3 h and up to 31.4 µL after 24 h. Although fluid uptake could not be quantified gravimetrically after 15 min of insertion into the phantom, slight diffusion of a dye previously incorporated into the hydrogel was observed visually, suggesting limited but detectable absorption. This limited uptake may be attributed to the swelling kinetics of Dex-MA hydrogels, which typically reach equilibrium within 1–2 h when fully immersed in saline solution [36]. However, even small amounts of fluid may be sufficient to carry out several analyses. For example, Steinbach et al. successfully quantified IgG and cytokine levels related to bovine response to tuberculin using ISF absorbed by a calcium-crosslinked alginate hydrogel layer coated onto stainless-steel solid microneedles [19]. However, the authors did not report the collected ISF volume, although it can be inferred that it was limited due to the small size of the hydrogel layer. In contrast, our design features microneedles composed entirely of hydrogel, potentially allowing for greater fluid uptake.

These preliminary findings indicate that Dex-MA microneedle arrays are capable of absorbing interstitial fluid upon insertion and suggest that a minimum application time of several hours may be required to achieve appreciable fluid uptake (tens of µL) in vivo.

## 5. Conclusions

Wearable sensors are urgently need for pets and farm animals to monitor physiology deviations in a continuous manner when facing specific challenges and environmental hazards, and to improve management. Regarding contact-sensing techniques, considerable challenges still need to be overcome, the first one being our capacity to design painless devices but robust enough to allow data acquisition in adverse housing environments.

To date, only a few studies have investigated microneedle-based devices for interstitial fluid sampling and analysis in animals. They relied on the use of stainless-steel hollow microneedles in rats [20], and hydrogel-coated 1.4 mm high stainless-steel solid microneedles for cows [19]. This study developed an original approach, based on MNAs comprising only crosslinked Dex-MA hydrogel, for improved fluid volume sampling, and the tallest microneedles, to overcome the challenges of higher animal skin thickness. We demonstrated the feasibility of using Dex-MA hydrogel-based MNAs to perforate the skin and give access to interstitial fluid in large animal models, specifically pigs and cattle.

We successfully developed a robust and reproducible fabrication protocol yielding MNAs with 2.8 mm-tall MNs—the tallest hydrogel-based structures of their kind reported to date, to our knowledge. Ex vivo investigations using porcine and bovine tissues showed the capability of the MNAs to perforate multiple anatomical sites effectively. Additionally, fluid absorption assays performed with a skin-mimicking hydrogel demonstrated that the MNAs could uptake between 10 and 30 µL of fluid within a few hours.

The experimental methodology was designed to closely replicate realistic application conditions, thereby laying the groundwork for subsequent in vivo studies. The results indicate that an application duration of several hours is required to achieve meaningful fluid uptake in the order of tens of microliters, and identify the neck as a particularly suitable anatomical site for secure and sustained MNA placement. Building upon these findings, future in vivo experiments are planned to refine application protocols, evaluate interstitial fluid uptake efficacy under physiological conditions, and facilitate translation to practical field deployment.

Overall, this work provides a critical step toward the development of advanced biosampling technologies for animal health monitoring, with potential applications in precision livestock farming and veterinary diagnostics.

## Figures and Tables

**Figure 1 micromachines-16-01015-f001:**
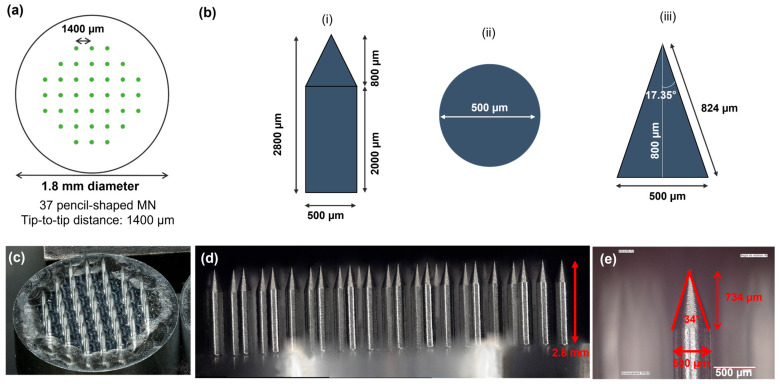
Design and digital images of the aluminum master featuring the microneedle array (MNA) geometry. (**a**) Top view of the MN array design. (**b**) Pencil-shaped MN design: (i) side view; (ii) top view; (iii) detail of the MN tip geometry. (**c**) Overview of the 18 mm diameter MNA with 37 microneedles (MN). (**d**) Side image of the MNA. (**e**) Detail of the geometry of a MN tip.

**Figure 2 micromachines-16-01015-f002:**
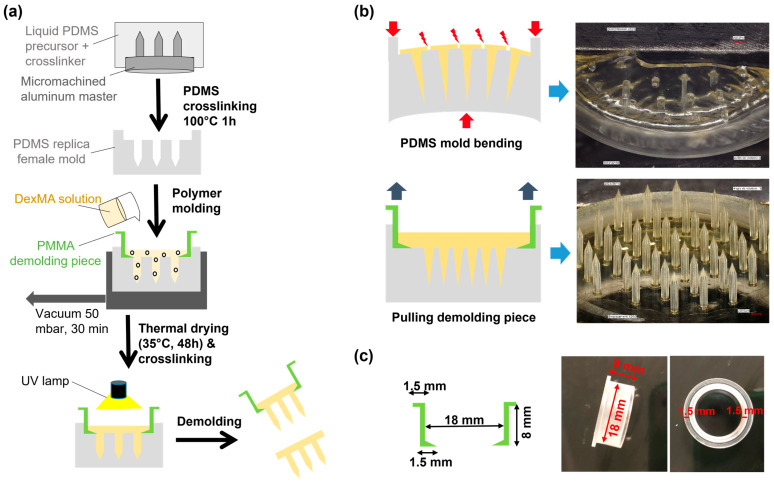
Fabrication of dextran-methacrylate (Dex-MA) MNAs with high MNs. (**a**) Schematic flow of the fabrication process with the preparation of the polydimethylsiloxane (PDMS) mold, polymer molding, drying, UV crosslinking, and demolding. (**b**) Details of the demolding step without (top images) and with (bottom images) a poly(methyl methacrylate) (PMMA) demolding piece. (**c**) Design and photographs of the PMMA demolding piece.

**Figure 3 micromachines-16-01015-f003:**
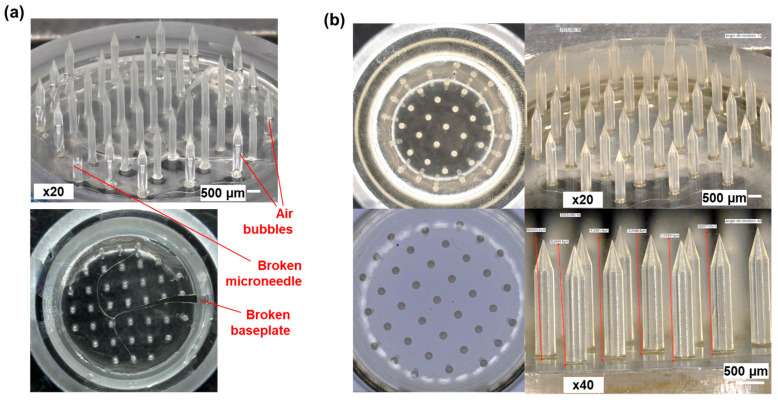
Optimization of the polymer formulation for the fabrication of Dex-MA MNAs with high microneedles. (**a**) Molding of MNAs using Dex-MA/LAP formulation (Dex-MA 20% *w*/*v*, LAP 1% *w*/*v*, in distillated water). (**b**) Molding of MNAs using Dex-MA/LAP/sorbitol formulation (Dex-MA 20% *w*/*v*, LAP 1% *w*/*v*, sorbitol 5% *w*/*v*, in distillated water).

**Figure 4 micromachines-16-01015-f004:**
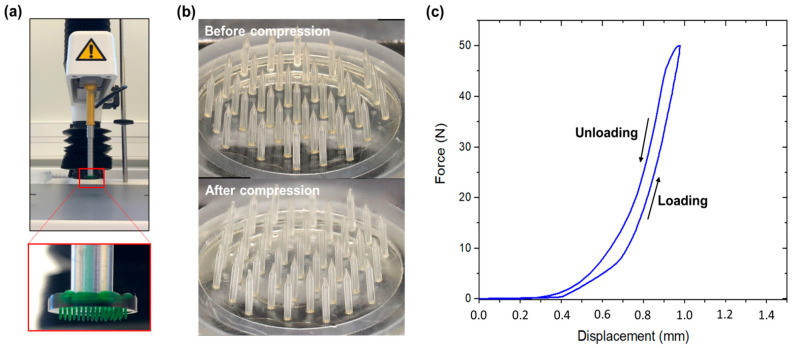
Mechanical characterization of Dex-MA MNAs with high microneedles. (**a**) Photograph of the experiment test with the MNA attached to the texturometer probe being compressed onto a steel plate. (**b**) Digital microscope images of a MNA before and after compression. (**c**) Force = f(displacement) curve during loading and unloading.

**Figure 5 micromachines-16-01015-f005:**
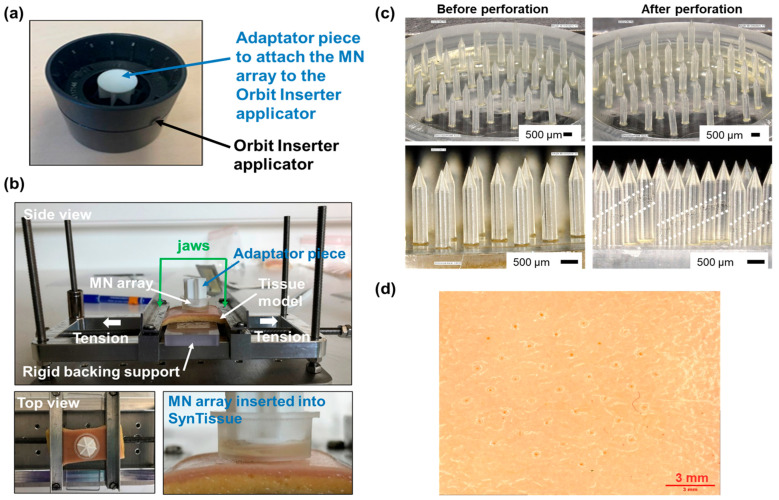
Perforation of human skin model. (**a**) Photograph of the Orbit Inserter assembled with the adaptation piece to attach the MNA. (**b**) Photographs of the perforation bench mounted with SynTissue after application of an MNA. (**c**). Digital microscopy images of a Dex-MA MNA before and after 2 min insertion in SynTissue. (**d**) Digital microscopy image of SynTissue after the insertion of a Dex-MA MNA.

**Figure 6 micromachines-16-01015-f006:**
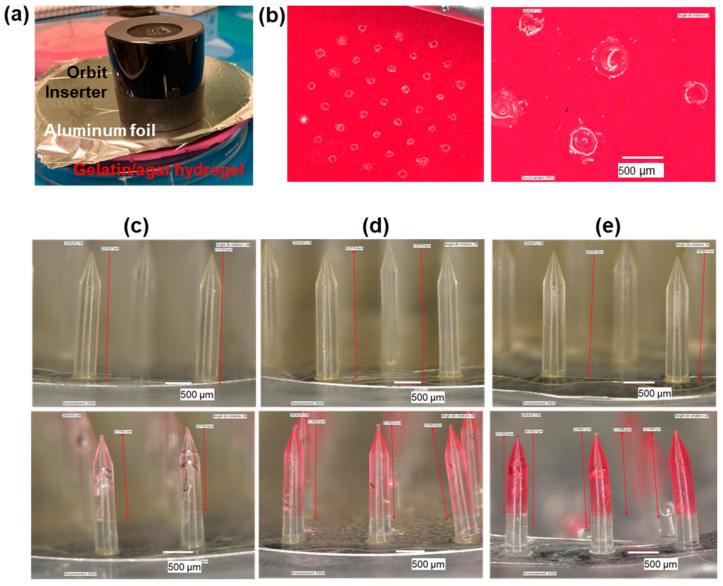
Fluid uptake in gelatin/agar hydrogel model. (**a**) Photograph of the insertion of a MNA into the skin phantom with the use of the Orbit Inserter. (**b**) Digital microscopy images of the gelatin/agar hydrogel after 15 min application of an MNA. (**c**–**e**) Digital microscopy images of the MNAs before (**top**) and after (**bottom**) insertion into and retrieval from the gelatin/agar hydrogel after 15 min (**c**), 3 h (**d**), and 24 h (**e**) of application.

**Figure 7 micromachines-16-01015-f007:**
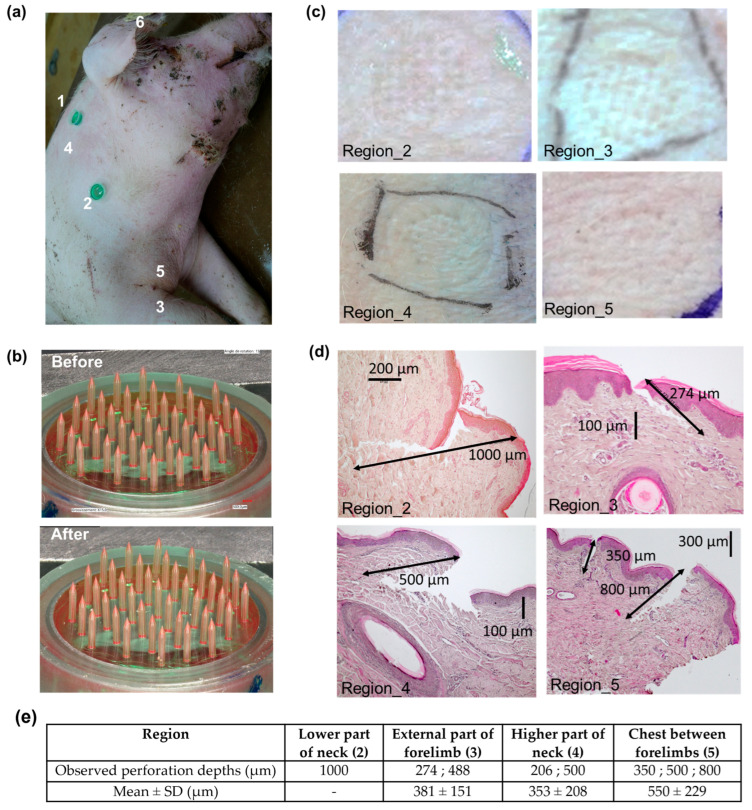
Porcine skin perforation experiments. (**a**) Photograph of the animal and the different regions where perforation was tested: right side of the neck behind the ear (1), lower part of the neck (2), external part of the right forelimb (3), higher part of the neck (4), chest between the two forelimbs (5), and inner surface of the ear (6). (**b**) Digital microscopy images of a MNA before and after insertion. (**c**) Photographs of different regions of skin after MNA removal. (**d**) Histology images of skin samples collected from different regions after MNA insertion. (**e**) Perforation depth measurements issued from histology images.

**Figure 8 micromachines-16-01015-f008:**
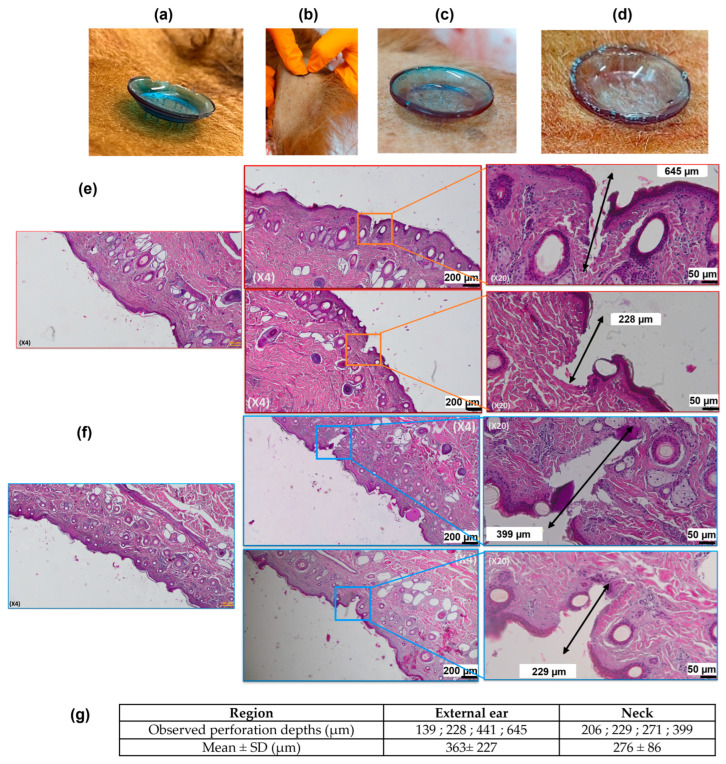
Cow skin perforation experiments. (**a**) Photograph of a MNA applied with the Orbit Inserter applicator on cow’s neck tissue. (**b**) Photograph of the application of a MNA on cow’s ear tissue by thumb pressure by exerting a counter-push. (**c**,**d**) Photographs of MNAs applied by thumb pressure on cow’s external ear (**c**) and neck (**d**) tissues. (**e**,**f**) Histology images of cow’s external ear (**e**) and neck (**f**) tissues: (i) non-perforated tissue region; (ii, iii): perforated tissue regions at ×4 (left) and ×20 (right) magnifications. (**g**) Perforation depth measurements issued from histology images.

**Table 1 micromachines-16-01015-t001:** Fluid diffusion from the gelatin/agar hydrogel to the MNA.

Duration of MNA Insertion	Amount of Up Taken Fluid (mg)	Length of MN Tip into Which the Liquid Has Diffused (µm) (% of the MN Total Height)
15 min	ND ^1^	1665 (65%)
3 h	10.7	1908 (75%)
24 h	31.4	1579 (62%)

^1^ ND: not measurable.

## Data Availability

The raw data supporting the conclusions of this article will be made available by the authors on request.

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
