# Peer review of "Design of Hydrogel Microneedle Arrays for Physiology Monitoring of Farm Animals"

_micromachines, 2025, doi:10.3390/mi16091015_

Round 1
Reviewer 1 Report
Comments and Suggestions for Authors
Design of hydrogel microneedle arrays for interstitial fluid analysis and animal physiology monitoring: Micromachines-3808125
General Concept Comments:
This research study demonstrates the feasibility of using photo-crosslinked Dex-MA hydrogel-based MNAs to perforate the skin and give access to interstitial fluid in large animal models, specifically pigs and cattle. The study reported fabrication of hydrogel MNs with a height of 2.8 mm-tall, mechanical testing, perforation studies on synthetic and ex vivo skin models, and fluid uptake experiments. Ex vivo investigations using porcine and bovine tissues demonstrated the capability of the MNAs to perforate multiple anatomical sites. Fluid absorption assays demonstrated 10 and 30 μL of fluid uptake within a few hours
The manuscript demonstrates a clear technical novelty and was presented in a well-structured manner. With major revisions to enhance clarity and rigor, the research study would be a valuable contribution to the development of advanced biosampling technologies for animal health monitoring.
Specific Comments:
- The authors claims that the hydrogel microneedles with 2.8 mm are the tallest one fabrication so far, I suggest benchmarking the results with the prior literature interms of materials, dimensions and mechanical properties.
- The choice of height for the MNs was selected based on the thickness of the skin data, but can you also correlate it with respect to ISF target depths and biomarker accessibility at various anatomical sites in animals?
- I suggest including the CAD design of the aluminum master mold featuring the microneedle array (MNA) and the dimensions of the PMMA demolding tool design schematic as it would be valuable for reproducibility for readers.
- For borvin tissue manual thumb pressure was utilized instead of the orbit inserter. This rises the question related to the variability of the MNs penetration into the skin. Can authors provide more information on how this issue can be accounted for when applied for large animals?
- The authors reported the fluid uptake volume between 10–30 µL without any biomarket relevance. I suggest including the preliminary ISF analyte data related to the intended downstream assays such as ELISA, metabolomics, protein, etc..
- Compression testing against a rigid steel plate does not replicate the anisotropic and viscoelastic nature of animal skin. How can the compression testing be correlated to the actual skin surfaces ?. Fatigue or repeated-use tests were not discussed, which are crucial for considering multi-use molds or repeated applications.
- The MNs penetration depth was only reported from histology images. I suggest including statistical distribution by providing penetration depth boxplots with n-values for each anatomical site for both species.
- I suggest including the histological figures with high clarity and contrast with a scale bar to better visualize the MNs penetration.
Author Response
General Concept Comments:
This research study demonstrates the feasibility of using photo-crosslinked Dex-MA hydrogel-based MNAs to perforate the skin and give access to interstitial fluid in large animal models, specifically pigs and cattle. The study reported fabrication of hydrogel MNs with a height of 2.8 mm-tall, mechanical testing, perforation studies on synthetic and ex vivo skin models, and fluid uptake experiments. Ex vivo investigations using porcine and bovine tissues demonstrated the capability of the MNAs to perforate multiple anatomical sites. Fluid absorption assays demonstrated 10 and 30 μL of fluid uptake within a few hours
The manuscript demonstrates a clear technical novelty and was presented in a well-structured manner. With major revisions to enhance clarity and rigor, the research study would be a valuable contribution to the development of advanced biosampling technologies for animal health monitoring.
REP: We appreciate this positive comment. In the response to comments, all lines numbers refer to the ones of the revised manuscript with changes highlighted in red.
Specific Comments:
- The authors claims that the hydrogel microneedles with 2.8 mm are the tallest one fabrication so far, I suggest benchmarking the results with the prior literature in terms of materials, dimensions and mechanical properties.
REP: For humans, pain is observed for the insertion of MN-based devices which microneedles display height above 1500 µm (Gill, et al., Effect of Microneedle Design on Pain in Human Volunteers. The Clinical Journal of Pain 24(7):p 585-594, September 2008, DOI: 10.1097/AJP.0b013e31816778f9). Therefore, all studies we are aware of for human use describe microneedles with maximum 1.5-2 mm height (Ingrole et al., Trends of microneedle technology in the scientific literature, patents, clinical trials and internet activity. Biomaterials. 2021 Jan;267:120491. doi: 10.1016/j.biomaterials.2020.120491); above this value, the interest of MN-based devices as painless minimally invasive tools would be lost. For veterinary applications, the few references we identified (ref. [15-20] + a few less relevant ones) always described arrays with MN height of maximum 1500 µm, and a large majority (for drug or vaccine delivery, or for rodent studies, especially) around 600-650 µm tall. Our group previously fabricated 2.9 µm hollow MN arrays ([33], Simon, et al., Determination of an Implantation Area for Interstitial Fluid Extraction in Cows and Feasibility of Adapted Microneedles. Biosyst. Eng. 2022, 222, 62–70, doi:10.1016/j.biosystemseng.2022.07.007), but through the use of 3D printing for which the challenge remains miniaturization, and not through a polymer molding/evaporation process, for which obtaining tall MNs (and not small ones) is the challenge.
We added some benchmarks in lines 464- 467, modulated our statement and changed accordingly the sentence lines 482-484 so that it clearly specifies the type of MN material (hydrogel-based) and the process (polymer molding/evaporation process).
2. The choice of height for the MNs was selected based on the thickness of the skin data, but can you also correlate it with respect to ISF target depths and biomarker accessibility at various anatomical sites in animals?
REP: We thank the reviewer for this very relevant comment that to our knowledge is scarcely discussed in the literature, even for humans. Most of the MN and ISF literature generally consider that ISF is a “blood filtrate of plasma” and mainly highlight the difference of composition that can exist between the two fluids, especially for proteins with molecular weight above 70,000 kDa, and the lag time between ISF and blood concentrations. Comparison has been notably made between two ISF collection sites, dermal or adipose layer, for glucose monitoring (e.g. Thennadil et al., Comparison of glucose concentration in interstitial fluid, and capillary and venous blood during rapid changes in blood glucose levels. Diabetes Technol Ther. 2001 Fall;3(3):357-65. doi: 10.1089/15209150152607132.). Dermis is also outlined as the most relevant compartment for ISF uptake because of its structure and organization which presents low vascularization and low cell density, in contrast to epidermis and hypodermis. Other studies deal with the comparison of ISF composition when sampling using different methods (MNs, microdialysis, etc.). However, it does not seem that the potential differences of composition between ISF samples up taken at different skin depths have been extensively discussed, though Young et al. noted that “if the measurement goal is not just blood correlation but rather tissue penetration of a protein” (for instance after topical application ?), hypodermis could not be inferior to the dermis as a compartment to sample ISF (Young et al., Perspective-The Feasibility of Continuous Protein Monitoring in Interstitial Fluid. ECS Sens Plus. 2023 Jun 1;2(2):027001. doi: 10.1149/2754-2726/accd7e).
In animals, the very scarce literature on ISF sampling (lines 69-75) have not investigated that point either, nor the composition of ISF at different anatomical locations, to the extent of our knowledge.
We however noted the recent work of Xiong et al. showing that the ISF sampling method could impact the obtained metabolite concentrations (in their case, cations) (Xiong et al., Comparison of three methods for collecting interstitial fluid from subcutaneous tissue in mini pigs, MethodsX 12, 102700, (2024), DOI:10.1016/j.mex.2024.102700). In our study, we primarily considered anatomical regions that would be compatible with a sensor being worn by an animal moving around in its housing environment. To note that Steinbach et al. also used the side of cow neck to apply their MN-based ISF sampling device to assess the animal response to tuberculin [19] (addition of this remark in lines 511-512). This location constraint primarily determined the thickness of the skin that would be considered for designing the MNs. This was added in the revised version (lines 442-444)
Altogether, much work is still needed to better understand the ISF composition and its correlation to blood, its variation composition according to skin depth and anatomical locations, and identify the most suited method to use to sample it according to the targeted biomarker to analyze. The development of minimally invasive sampling tools such as the hydrogel-based MN arrays described in our study could facilitate and promote such investigation in the future to answer to these questions.
3. I suggest including the CAD design of the aluminum master mold featuring the microneedle array (MNA) and the dimensions of the PMMA demolding tool design schematic as it would be valuable for reproducibility for readers.
REP: We added more details on the design of the aluminum master mold described in Figure 1. We added more details on the design of the PMMA demolding tool in Figure 2. Figure legends were modified accordingly.
4. For borvin tissue manual thumb pressure was utilized instead of the orbit inserter. This rises the question related to the variability of the MNs penetration into the skin. Can authors provide more information on how this issue can be accounted for when applied for large animals?
REP: Whereas for pigs the commercial Orbit inserter applicator was sufficient to perforate the skin and its use provided reduced variability on the MN array application protocol, this type of applicator was indeed not efficient to perforate bovine skin. However, the use of thumb pressure allowed us to demonstrate that bovine skin perforation was possible, and to validate the overall concept of the hydrogel-based MN array to access bovine ISF, which was the main objective in this paper. To address the challenge of reproducible MN array application into cattle skin dermis, we are presently developing a more efficient home-made applicator, i.e., for which application force and speed of application can be tuned, specifically to address bovine skin perforation. This home-made applicator will be described and used in a follow-up study.
5. The authors reported the fluid uptake volume between 10–30 µL without any biomarket relevance. I suggest including the preliminary ISF analyte data related to the intended downstream assays such as ELISA, metabolomics, protein, etc..
REP: The present paper is really focused on the preliminary validation of the proof-of-concept, addressing the challenges of the fabrication of tall hydrogel-based MN necessary for farm animals, especially cattle and pigs, and demonstrating the ability of such devices to perforate the skin. Following the rules of 3Rs (regulation of ethical use of animals in product testing), the skin samples were obtained post mortem from healthy freshly euthanized animals. Fluid circulation ceases following the loss of cardiac activity. Therefore, we did not intend to assess and analyze fluid absorption in these euthanized animals, and remove the device after short application times (10-20 minutes for pig, 1-2 minutes for cow). We conducted fluid sampling studies under well (best) controlled conditions, using a gel tissue model, in which we could collect 10-30 µL of fluid after a few hours.
The scope of the present study was clarified in the introduction (lines 103-113) and the procedure was discussed lines 530-538.
6. Compression testing against a rigid steel plate does not replicate the anisotropic and viscoelastic nature of animal skin. How can the compression testing be correlated to the actual skin surfaces ? Fatigue or repeated-use tests were not discussed, which are crucial for considering multi-use molds or repeated applications.
REP: The objective of the compression test described in section 3.2 is merely to have a qualification of the mechanical toughness and behavior of the polymer alone and when shaped with the MN design, but for sure not to represent an interaction with skin surfaces. We are routinely using this test in the lab to compare different materials or MN designs. For instance, we initially investigated another shape for the MN, a conic shape, but this test showed us that the pencil-shape was far more resistant to compression than the conic one that was abandoned. To mimic (on an elementary basis) a perforation assay/an interaction with skin surface “in vitro”, we used the SynTissue model as described in section 3.3. To ensure that compression is sufficient to allow the MN to perforate a “real” skin, we performed the assays on the animals to include the viscoelastic nature of the skin in two species.
Fatigue or repeated use tests were not performed since these hydrogel-based MN arrays are intended only for single use: they are not expensive, and multiple use in the same or different animals appear to us not safe and will provoke cross-contaminations which are not acceptable for the ethics in animals used for scientific purposes (rules of 3Rs), as well as for the management of a farm (biosecurity rules). A same PDMS mold can be used several times to obtain series of devices; in the lab, we control all MN arrays after demolding, and even if the fidelity of the devices to the master aluminum mold is very high after 5-6 uses of a same PDMS mold (>95% or more), we systematically change the PDMS molds after the fabrication of 5 batches of devices to ensure high quality production of MN arrays.
7. The MNs penetration depth was only reported from histology images. I suggest including statistical distribution by providing penetration depth boxplots with n-values for each anatomical site for both species.
REP: Due to the difficulty of the preparation of tissues for histology and for sectioning the tissue where perforations can be observed, we do not have many data per location in the two animal species. Therefore, it is difficult to derive a strong statistical analysis from the data, just general tendency. However, we added in Figures 7 and 8, a table summarizing the data we have.
For porcine model, only locations 2-5 could be analyzed by histology since an issue was encountered during the preparation of the tissue sample coming from location 1. Data are:
|
Region |
Lower part of neck (2) |
External part of forelimb (3) |
Higher part of neck (4) |
Chest between forelimbs (5) |
|
Observed perforation depths (µm) |
1000 |
274 - 488 |
206 - 500 |
350 - 500 - 800 |
|
Mean ± SD (µm) |
- |
381 ± 151 |
353 ± 208 |
550 ± 229 |
For cow model, we completed data analysis with additional samples and conclusions were consequently modified (lines 385-386).
|
Region |
External ear |
Neck |
|
Observed perforation depths (µm) |
139-228-441-645 |
206-229-271-399 |
|
Mean ± SD (µm) |
363± 227 |
276 ± 86 |
8. I suggest including the histological figures with high clarity and contrast with a scale bar to better visualize the MNs penetration.
REP: Histological images are original images without contrast change, with just enhancement of scale bars, and we prefer conserving original images. The scale bars of Figure 8 were indeed not yet enhanced, and we have made the modification to make them more apparent for the reader.
Reviewer 2 Report
Comments and Suggestions for Authors
This paper investigates using microneedle arrays to extract interstitial fluid from farm animals that require longer/deeper penetration into the skin. The paper is generally well written, although the organization can be improved for better readability.
-The scope of the study can be better described in the last paragraph of the Introduction section.
-It would be useful to summarize the dimension differences between humans and animals in a table for easier comparison.
-Include some reasoning about why choosing dextran-methacrylate as the material
-Some contents in Sec. 3 are more suitable for the Method section.
-Differentiate loading and unloading curves in Fig. 4(c) with color or line style.
-What amount of fluid uptake will be clinically relevant or required for analysis?
-Articulate the innovation of this work more clearly.
Author Response
This paper investigates using microneedle arrays to extract interstitial fluid from farm animals that require longer/deeper penetration into the skin. The paper is generally well written, although the organization can be improved for better readability.
REP: We thank the reviewer for this positive comment and we carefully addressed all the comments detailed below.
In the response to comments, all lines numbers refer to the ones of the revised manuscript with changes highlighted in red.
-The scope of the study can be better described in the last paragraph of the Introduction section.
REP: We modified the last paragraph of the introduction (lines 103-111) to better enhance the scopes of the study: 1) overcoming the technological challenges of the fabrication of hydrogel-based MN arrays with tall microneedles using a polymer molding/evaporation process, and 2) establishing the proof-of-concept that such hydrogel-based MN arrays are able to perforate animal farm skin with the goal of sampling ISF to monitor animal physiology.
-It would be useful to summarize the dimension differences between humans and animals in a table for easier comparison.
REP: The comparison between human, porcine, and cattle skin properties issued from literature are detailed in lines 91-99.
For human/pig comparison, we attached to data issued from the same publications making direct comparison between the two species ([28] for Young’s modulus, [30,31] for the thickness of the epidermis-stratum corneum layers and full skin) since a wide range of variability in such data can be found across different publications, especially for Young’s modulus measurements, inherent to skin tissue variability between ages, body location…, and according to the measurement protocols.
For cow skin properties, unfortunately, the data are very scarce, and we cited the few references we could find on the topic. Data on ear skin thickness seem reliable and our group performed its own study on the topic ([33] Simon et al., Determination of an Implantation Area for Interstitial Fluid Extraction in Cows and Feasibility of Adapted Microneedles. Biosyst. Eng. 2022, 222, 62–70, doi:10.1016/j.biosystemseng.2022.07.007). However, for cow skin Young’s modulus, no data was found (there exist few data on leather properties but in this situation, the skin has been dried and submitted to various chemical treatments, making these data surely over-estimated in comparison to hydrated alive skin and therefore not relevant). The closest value was the one reported for films of collagen extracted from cattle’s skin [29].
Since all the data we cited are issued from literature (except our previous study on cow ear skin thickness [33]) and were not measured directly in this work, and considering that some of them should be taken cautiously, especially concerning cow skin Young’s modulus, we feel that presenting figures in a Table could mislead the readers and this could also encourage biased citations in the next years. Note that the indicated values should be more considered as orders of magnitude and on a comparative basis than absolute.
-Include some reasoning about why choosing dextran-methacrylate as the material
REP: We thank the reviewer for this very valuable comment. The information was indeed lacking and we added in the last paragraph of the introduction section (lines 109-113) the justification of material selection, that is based on our previous work (refs. [36] and [41], Darmau et al., Water-Based Synthesis of Dextran-Methacrylate and Its Use to Design Hydrogels for Biomedical Applications. Eur. Polym. J. 2024, 221, 113515, doi:10.1016/j.eurpolymj.2024.113515 and Darmau et al., Self‐Extracting Dextran‐Based Hydrogel Microneedle Arrays with an Interpenetrating Bioelectroenzymatic Sensor for Transdermal Monitoring with Matrix Protection. Adv. Healthc. Mater. 2025, 14, doi:10.1002/adhm.202403209).
-Some contents in Sec. 3 are more suitable for the Method section.
REP: We removed from section 3 lines 302-305 and slightly modified lines 182-183 in the Method section 2. We removed from section 3 lines 329-332, 334-335, and slightly modified line 195 in the Method section 2.
-Differentiate loading and unloading curves in Fig. 4(c) with color or line style.
REP: We specified the loading and unloading part of the curve in Figure 4c.
-What amount of fluid uptake will be clinically relevant or required for analysis?
REP: In human, ISF sampling volumes (through MN devices or other tools) usually vary between a few to a few tens of µL (1-60 µL) ([7-14]). These quantities have been shown relevant for the analysis of antibodies, cytokines, etc., moreover that analysis can be carried out after dilution of the ISF sample. Provided that the concentration of these metabolites are similar in the ISF of farm animals as in the ISF of human, it should be possible to make their analysis starting from a few tens of µL (typically 10-20 µL).
We also added a comment on ref [19] (lines 544-551): “However, even small amounts of fluid may be sufficient to carry out several analyses. For example, Steinbach et al. successfully quantified IgG and cytokine levels related to bovine response to tuberculin using ISF absorbed by a calcium-crosslinked alginate hydrogel layer coated onto stainless-steel solid microneedles [19]. However, the authors did not report the collected ISF volume, although it can be inferred that it was limited due to the small size of the hydrogel layer. In contrast, our design features microneedles composed entirely of hydrogel, potentially allowing for greater fluid up-take”.
-Articulate the innovation of this work more clearly.
REP: The last paragraph of the introduction section was reviewed and better position the novelty of the work (lines 103-113) in comparison to previous literature on MN-based arrays for ISF uptake in animal detailed in lines 69-75.
The discussion section highlights the technical novelty in the fabrication process (section 4.3, particularly lines 482-484: “These protocol enhancements enabled the successful fabrication of MNAs with 2.8 mm-high MNs—the tallest hydrogel-based microneedles reported to date fabricated through a polymer molding/evaporation process, to the best of our knowledge.”), as well as the conclusion in lines 570-572.
We added a few lines at the beginning of the conclusion (lines 557-567) to better enhance the work novelty. We underlined also that very few devices target farm animals with anatomical and behavioral specificities that complicate the design of microneedles.
Reviewer 3 Report
Comments and Suggestions for Authors
This article presents a hydrogel-forming microneedle array for interstitial fluid sampling. The authors opted for relatively long needles to accommodate anatomical differences in skin thickness between humans and large farm animals. i have several key concern:
1. A 2.8 mm needle likely reaches past the epidermis into the deep dermis, where pain-sensitive nerve endings reside. This could cause significant discomfort. Additionally, deeper penetration increases the risk of rupturing dermal capillaries, potentially contaminating ISF samples and compromising analytical accuracy. Effective MN design typically limits needle length to a few hundred micrometers to avoid such issues.
2. The current study does not include any actual ISF composition analysis or physiological monitoring to validate sampling efficacy. Thus, labeling the work primarily as an “ISF sampling” study may be premature without experimental proof of fluid collection and analyte integrity.
3. From Figures 6c–d, the hydrogel-forming microneedles exhibit minimal morphological change after 24 h incubation in a 75% water agarose gel, suggesting a very high crosslinking density. However, this parameter isn’t quantified or characterized. Clarifying the hydrogel’s crosslinking degree and relating it to swelling behavior and mechanical properties would enhance interpretability and reliability.
Author Response
In the response to comments, all lines numbers refer to the ones of the revised manuscript with changes highlighted in red.
- A 2.8 mm needle likely reaches past the epidermis into the deep dermis, where pain-sensitive nerve endings reside. This could cause significant discomfort. Additionally, deeper penetration increases the risk of rupturing dermal capillaries, potentially contaminating ISF samples and compromising analytical accuracy. Effective MN design typically limits needle length to a few hundred micrometers to avoid such issues.
REP: We selected high MNs based on the analysis of animal farm skin thickness, especially for cows, that is largely superior to humans (lines 96-99: thickness of epidermis and stratum corneum layers: ≈ 55-90 µm for pig [30–32], 600-1400 µm for ear cow [33], in comparison to 50-60 µm for humans ([30,31]; full skin thickness: 1.3 mm (ear)-3.6 mm (back) for pig [30–32], 1.4-3.2 mm for ear cow [33], 3-8 mm for other cow regions [34,35], in comparison to 1.5 mm for human forearm ([30,31]). Especially, we preferred to target values in the upper range so that we were able to test different body locations, especially for cattle. In fact, we qualitatively tested in a very preliminary experiment a different design with shorter MNs in cow (1.4 mm MN length), and no perforation was observed by thumb pressure. We therefore really focused further experiments on the 2.8 mm design, for which we could observe perforation and MN insertion.
Moreover, it has to be taken into account that in many cases, and especially for hydrogel-based MNs that are not as tough as metal or silicon ones, not all the MN length penetrates the skin, but it is generally acknowledged that in human for usual MN array designs and quite efficient insertion, about only 60% of the total MN length penetrates the tissue ([45] Chua et al., Effect of Microneedles Shape on Skin Penetration and Minimally Invasive Continuous Glucose Monitoring in Vivo. Sens. Actuators Phys. 2013, 203, 373–381, doi:10.1016/j.sna.2013.09.026.), though this value is highly dependent for a defined material of the MN shape, MN density, and MN length. Higher MNs tends to improve penetration depth (Makvandi et al. Nano-Micro Lett 13, 93 (2021)). We achieved 60-75% of MN length penetration in the SynTissue model, demonstrating that our design is competitive to the ones described in the literature.
We observed perforation depths of about 200-1000 µm in pork skin, and 140-650 µm in cow skin, which are values far below the depth at which discomfort or bleeding would be observed. Additionally, we are now conducting tests of tolerance of the design when placed on the animals for several hours. We did not notice any warning reactions neither on the behavior nor on the skin appearance when the device was removed, which make us confident for the next steps. Histological images confirmed that the device insertion was only into the dermis compartment. These low values were discussed in the section 4.4, with possible reasons: under-estimation due to histological sample preparation, increased tissue stiffness post-mortem, and for sure reduced penetration efficiency into ex vivo animal skin in comparison to model human skin because of higher mechanical properties. It is however true that the perforation efficacy might eventually be improved by further optimizing the MN array designs (MN shape, length, density, etc…). We added a sentence in the discussion lines 526-528: “Future studies exploring various design parameters (e.g., microneedle shape, length, and density) could also enable greater or more consistent perforation depths.”
- The current study does not include any actual ISF composition analysis or physiological monitoring to validate sampling efficacy. Thus, labeling the work primarily as an “ISF sampling” study may be premature without experimental proof of fluid collection and analyte integrity.
REP: The present paper is really focused on the preliminary validation of the proof-of-concept, addressing the challenges of the fabrication of tall hydrogel-based MN necessary for farm animals, especially cattle and pigs, and demonstrating the ability of such devices to perforate the skin. Following the rules of 3Rs (regulation of ethical use of animals in product testing), the skin samples were obtained post mortem from healthy freshly euthanized animals. Fluid circulation ceases following the loss of cardiac activity. Therefore, we did not intend to assess and analyze fluid absorption in these euthanized animals, and remove the device after short application times (10-20 minutes for pig, 1-2 minutes for cow). We conducted swelling studies under well (best) controlled conditions using a gel tissue model, in which we could collect 10-30 µL of fluid after a few hours. We now clarify this point in lines 530-536.
We fully understand the reviewer’s comment and modify the article title accordingly; the new title is “Design of hydrogel microneedle arrays for physiology monitoring of farm animals”.
- From Figures 6c–d, the hydrogel-forming microneedles exhibit minimal morphological change after 24 h incubation in a 75% water agarose gel, suggesting a very high crosslinking density. However, this parameter isn’t quantified or characterized. Clarifying the hydrogel’s crosslinking degree and relating it to swelling behavior and mechanical properties would enhance interpretability and reliability.
REP: In fact, very slight geometrical modifications of the MNs after fluid sampling exist, but are really minimal (a few % increase of MN diameter) and not easily observable, as we underlined in our previous publications (refs. [36] and [41], Darmau et al., Water-Based Synthesis of Dextran-Methacrylate and Its Use to Design Hydrogels for Biomedical Applications. Eur. Polym. J. 2024, 221, 113515, doi:10.1016/j.eurpolymj.2024.113515 and Darmau et al., Self‐Extracting Dextran‐Based Hydrogel Microneedle Arrays with an Interpenetrating Bioelectroenzymatic Sensor for Transdermal Monitoring with Matrix Protection. Adv. Healthc. Mater. 2025, 14, doi:10.1002/adhm.202403209). These slight geometrical modifications combined with the sampling capability of the MN is a positive feature, since devices with highly swelling material and large MN diameter increase could easily break during device retrieval from the skin.
The Dex-MA material properties including Young’s modulus and equilibrium swelling ratio were already detailed in discussion section 4.1 (lines 416-421). This important information justifying the Dex-MA hydrogel selection was also briefly added in the last paragraph of the introduction section (lines 109-113).
Round 2
Reviewer 1 Report
Comments and Suggestions for Authors
All the suggested recommendations were addresses.
The research study would be a valuable contribution to the development of advanced biosampling technologies for animal health monitoring.
Reviewer 3 Report
Comments and Suggestions for Authors
The authors have addressed all my concerns. It could be published in its current status.